# HIV-1 Vpr suppresses expression of the thiazide-sensitive sodium chloride co-transporter in the distal convoluted tubule

Shashi Shrivastav[1ʘ], Hewang Lee[1,2ʘ], Koji Okamoto[1ʘ¤a], Huiyan Lu[1ʘ], Teruhiko Yoshida[1], Khun Zaw Latt[1], Hidefumi Wakashin[1], James L. T. Dalgleish[1¤b], Erik H. Koritzinsky[1¤c], Peng Xu[3], Laureano D. Asico[2], Joon-Yong Chung[4], Stephen Hewitt[4], John J. Gildea[3], Robin A. Felder[3], Pedro A. Jose[2‡], Avi Z. Rosenberg[5‡], Mark A. Knepper[6‡], Tomoshige Kino[7‡], Jeffrey B. Kopp[1‡*]

1 Kidney Disease Section, Kidney Diseases Branch, NIDDK, NIH, Bethesda, Maryland, United States of America, 2 Department of Medicine, The George Washington University School of Medicine & Health Sciences, Washington, DC, United States of America, 3 Department of Pathology, University of Virginia, Charlottesville, Virginia, United States of America, 4 Experimental Pathology Laboratory, Laboratory of Pathology, Center for Cancer Research, NCI, NIH, Bethesda, Maryland, United States of America, 5 Department of Pathology, Johns Hopkins Medical Institutions, Baltimore, Maryland, United States of America, 6 Epithelial Systems Biology Laboratory, Systems Biology Center, Division of Intramural Research, NHLBI, NIH, Bethesda, Maryland, United States of America, 7 Laboratory for Molecular and Genomic Endocrinology, Division of Translational Medicine, Sidra Medicine, Doha, Qatar

ʘ These authors contributed equally to this work.
¤a Current address: Division of Nephrology, Endocrinology and Vascular Medicine, Department of Medicine, Tohoku University Hospital, Aoba-ku, Sendai, Miyagi, Japan
¤b Current address: Department of Biostatistics, Columbia University, New York, New York, United States of America
¤c Current address: Renal Diagnostics and Therapeutics Unit, Kidney Diseases Branch, NIDDK, NIH, Bethesda, Maryland, United States of America
‡ These authors are joint senior authors on this work.
* jbkopp@nih.gov

**Data Availability Statement:** All relevant data are within the manuscript and its Supporting information files.

## Abstract

HIV-associated nephropathy (HIVAN) impairs functions of both glomeruli and tubules. Attention has been previously focused on the HIVAN glomerulopathy. Tubular injury has drawn increased attention because sodium wasting is common in hospitalized HIV/AIDS patients. We used viral protein R (Vpr)-transgenic mice to investigate the mechanisms whereby Vpr contributes to urinary sodium wasting. In phosphoenolpyruvate carboxykinase promoter-driven Vpr-transgenic mice, *in situ* hybridization showed that Vpr mRNA was expressed in all nephron segments, including the distal convoluted tubule. Vpr-transgenic mice, compared with wild-type littermates, markedly increased urinary sodium excretion, despite similar plasma renin activity and aldosterone levels. Kidneys from Vpr-transgenic mice also markedly reduced protein abundance of the $Na^+$-$Cl^-$ cotransporter (NCC), while mineralocorticoid receptor (MR) protein expression level was unchanged. In African green monkey kidney cells, Vpr abrogated the aldosterone-mediated stimulation of MR transcriptional activity. Gene expression of *Slc12a3* (NCC) in Vpr-transgenic mice was significantly lower compared with wild-type mice, assessed by both qRT-PCR and RNAScope *in situ* hybridization analysis. Chromatin immunoprecipitation assays identified multiple MR response elements (MRE), located from 5 kb upstream of the transcription start site and

**Funding:** This work was funded by the NIDDK Intramural Research Program 1ZIADK043411-15.

**Competing interests:** Author JBK holds a patent relating to monoclonal antibodies to HIV-1 Vpr and methods of using same. United States Patent 7,993,647 (2015). No other conflicts of interest, financial or otherwise, are declared by the authors. This does not alter our adherence to PLOS ONE policies on sharing data and materials.

extending to the third exon of the *SLC12A3* gene. Mutation of MRE and SP1 sites in the *SLC12A3* promoter region abrogated the transcriptional responses to aldosterone and Vpr, indicating that functional MRE and SP1 are required for the *SLC12A3* gene suppression in response to Vpr. Thus, Vpr attenuates MR transcriptional activity and inhibits *Slc12a3* transcription in the distal convoluted tubule and contributes to salt wasting in Vpr-transgenic mice.

## Introduction

HIV-AIDS remains an important global health problem. Individuals living with HIV-1 may manifest various forms of progressive chronic kidney disease (CKD), including HIV-associated nephropathy (HIVAN) [1,2]. The low viral load due to antiretroviral therapy effectively prolongs life but also results in chronic illness, affecting many organs, including tubular injury in the kidney [3,4].

Vpr, a small 96-amino acid HIV genome-encoded peptide, contributes to overall HIVAN pathogenesis in renal tubule epithelial cells. Chronically infected cells, including renal parenchymal cells, may continue to express Vpr even with antiretroviral therapy [5]. Through diverse interacting molecules [6], Vpr has multiple functions in the regulation of viral and host gene transcription [7]. Vpr binds to the transcription factor SP1 to stimulate the HIV-1 long terminal repeat to promote retroviral mRNA transcription in the early stage of infection [8]. Vpr also acts as an adapter, bridging promoter-bound transcription factors and the transcriptional coactivator p300/cAMP-responsive element-binding protein (CREB)-binding protein (CBP) to modulate host gene transcription [9].

Vpr is a coactivator of glucocorticoid-regulated gene transcription, leading to increased glucocorticoid sensitivity [10]. In preadipocytes and adipocytes, Vpr suppresses PPARγ-induced transactivation to inhibit cell differentiation [11], which may promote lipodystrophy [12]. In HepG2 cells, Vpr stimulates PPARβ/δ-induced transcriptional activity [13], leading to increased gene and protein expression of pyruvate dehydrogenase (PDH) kinase 4 [13] and subsequent inactivation of the PDH complex, which also contributes to renal tubular dysfunction [14].

The mineralocorticoid receptor (MR) is a member of the nuclear receptor superfamily that shares structural and functional similarity with the glucocorticoid receptor (GR) and regulates electrolyte balance, including sodium and potassium [15]. The MR plays an essential role in sodium reabsorption in the aldosterone-sensitive distal nephron, including the DCT the early or DCT1 and late or DCT2 sub-segments) [16]. The aldosterone-MR complex translocates from the cytoplasm to the nucleus and binds MR response elements (MREs) located in the promoters of aldosterone-responsive genes [15,17], where it regulates the transcriptional activity of these genes by attracting transcriptional cofactors, including SP1 [18].

Consistent with the tubular injury of HIVAN [3,4], it has been reported that sodium wasting is common in hospitalized HIV/AIDS patients in Chinese [19], European descent [20], and Indian [21] populations. However, the role of Vpr in the tubular histopathology of HIVAN is not well-characterized.

HIV patients receiving antiretroviral therapy have a low persistent Vpr expression in renal tubules. To elucidate the mechanism of renal sodium transport dysregulation observed in some HIV patients, we used a transgenic (Tg) mouse model that expressed low levels of Vpr in renal tubules. We tested the hypothesis that Vpr acts as a repressor of MR and thereby

promotes suppression of aldosterone-stimulated NCC gene expression in the distal nephron, contributing to renal salt wasting.

## Materials and methods

### Generation and maintenance of Vpr transgenic mice

The mouse experimental protocol was approved in advance by the NIDDK Animal Care and Use Committee. The generation, characterization, and genotyping of Tet-response element-driven Vpr-Tg mice have been described [22]. The phosphoenolpyruvate carboxykinase 1 (*Pepck1*) promoter was ligated onto tTA fragment, and the resulting transgene was co-injected with a tet-op/Vpr construct into mouse oocytes.

Tg and wild-type (WT) mice were maintained on a diet containing doxycycline (TD 98186, Envigo, Madison, WI) to inhibit the expression of Vpr. Mice were switched from doxycycline diet to 65% protein food (TD 190088, Envigo) for two weeks to induce transgene promoter activity. Then, the mice were fed with freshly prepared semi-solid sodium-deficient food [TD 90228 (3 g), containing casein (33.3%), agar (0.5%), sodium (0.045%), and 2.67 mL of water] for four days, to induce aldosterone activity. The mice were fed the measured amount of food for four days. We noted that there was no uneaten food in the cages. On day four, the mice were individually housed in metabolic cages to collect overnight urine samples. During the entire experimental period, the mice had *ad libitum* access to drinking water and were housed in a stress-free environment. The mice were anesthetized by the intraperitoneal (IP) injection of 0.3–0.5 ml of 2.5% Avertin (tert amyl alcohol, Sigma, St. Louis, MO) in 0.9% NaCl. After the anesthesia was in full effect, blood was withdrawn by cardiac puncture and the kidneys were harvested, leading to death due to exsanguination. To alleviate suffering, working Avertin solution was warmed to body temperature before IP injection; the mice were laid on a heating pad to keep the body temperature (37°C) stable during the entire surgical procedure. The mice were closely monitored throughout the procedure to ensure that the mice were humanely treated during the surgical and euthanasia procedures.

### Chromatin immunoprecipitation

Chromatin immunoprecipitation (ChIP) assay was performed using a commercial kit (Active Motif, Carlsbad, CA) [10,11,13]. Human distal convoluted tubule (DCT) cells [23] (60–70% confluent) were transfected for 12 hr with FLAG-MR-expressing plasmid, followed by exposure to sVpr (100 ng/mL) for 24 hr, with or without additional aldosterone (100 nM) exposure for one hr. Anti-FLAG antibody was used to immunoprecipitate MR-binding DNA fragments; recovered PCR DNA fragments (180–600 bps) were purified (Qiagen, Hilden, DE).

MRE candidate sequences within the promoter region of *SLC12A3* (from -5 kb to +3 kb relative to the transcriptional start site, NCBI reference sequence NM_000339.2) were identified, using Genomatix MatInspector (Genomatix, Ann Arbor, MI), in order to find candidate transcription factor binding sites. Candidate DNA sequences were compared against binding sites for transcription factors associated with glucocorticoid response element family for vertebrates (Genomatix identifier: V$GRE), using the search term "NR3C2" in Genomatix MatInspector, and with the negative MRE family (V$NGRE). S1 Table shows the coding sequences and matrices matching the potential candidate sequences.

We designed 15 primer pairs to amplify MRE DNA sequences (S2 Table). The threshold cycle (Ct) values of ChIP samples were normalized, using input DNA quantities. The relative amount of DNA precipitated by the anti-FLAG antibody, compared to that observed with negative control antibody, was expressed as fold immunoprecipitation.

## Mutagenesis analysis for MRE within human *SLC12A3* promoter

One putative MRE motif and one SP1 motif [24] in the proximal promoter region of the human *SLC12A3* gene were chosen for functional evaluation. The LightSwitch plasmid (SwitchGear Genomics, Carlsbad, CA) was used as a template. Mutations in the *SLC12A3* promoter were introduced using QuikChange Mutagenesis kit (Stratagene Cloning Systems, La Jolla, CA). Two potential motif sequences, sequences 5'-caatcaaatggTGTTCTgc-3' (amplicon P-9) and 5'- CCCTCCCTGGacacc-3' (amplicon P-10), in the native *SLC12A3* promoter plasmid pTSC-Luc were replaced with random sequences (S2 Table), which produced three constructs (mutated P-9, mutated P-10, and double mutants). Analysis of the random sequences by BLAST verified that these lacked functional motifs.

## Statistical analysis

Data are presented as mean ± standard deviation. Each dot in the graphs represents the average value of technical repeats of an individual mouse, and the "n" value in each group is the number of mice studied. Statistical significance was evaluated with Student's *t* test for two-paired groups and one-way ANOVA, with Bonferroni correction for multiple comparisons, using GraphPad Prism 7.0 software (GraphPad Software, La Jolla, CA). *P*-values less than 0.05 were considered statistically significant.

## Supporting methods

Complete methods, including DNA constructs, antibodies and reagents, chemical analysis, immunohistochemistry, single molecule RNA *in situ* hybridization, cell culture, transient transfection, and promoter reporter assays, western blot analysis, quantitative real-time polymerase chain reaction, and immunofluorescence microscopy are in **Supporting Methods**.

# Results

## Vpr mRNA and protein were expressed in the kidney of *Pepck* promoter-driven Tg mice

Under the control of the *Pepck1* promoter, Vpr is expressed in the liver and adipocytes in Tg mice, as previously reported [22], but its expression in the kidney has not been characterized. Using BaseScope *in situ* hybridization technology, Vpr mRNA expression was noted in all nephron segments, including glomeruli and various tubular segments (Fig 1). In the Vpr-Tg mice, Vpr protein was robustly expressed in the DCTs and other nephron segments in kidney sections observed by immunofluorescence staining (S1 Fig). These results indicate both Vpr mRNA and protein were expressed in the *Pepck* promoter-driven Tg mice.

## Vpr increased urinary sodium excretion in Vpr Tg mice

Compared with WT littermates, Vpr Tg mice had increased urinary sodium excretion (UNaV), both before and after salt depletion (Fig 2A). The fractional excretion of sodium (FENa) before and after salt depletion was also higher in Tg mice than WT mice (Fig 2B). Serum Na$^+$, K$^+$, and other electrolyte concentrations were similar in Tg, and WT mice fed salt-depleted diet (S2 Fig).

There were no differences in serum creatinine (Fig 2C) and creatinine clearance (Fig 2D) among the four groups of Tg and WT mice, with and without salt depletion, suggesting similar glomerular filtration rates. Blood urea nitrogen (BUN) concentrations were higher in Tg mice than WT mice, which were increased further by salt depletion (Fig 2E). The increase in UNaV

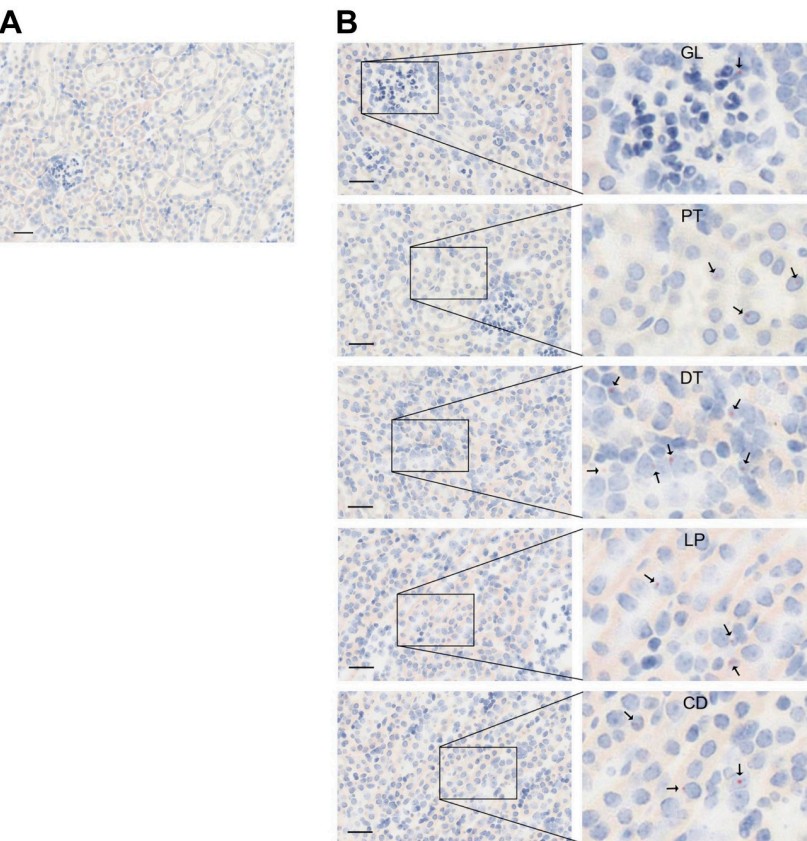

**Fig 1. Vpr gene expression in nephrons of Vpr-Tg mice analyzed by BaseScope.** (**A**) negative control image from the WT mouse kidney section which was hybridized using a Vpr probe. (**B**) The same Vpr probe was hybridized as (**A**) Vpr RNA (pink), indicated by arrows in images from Tg mouse kidney tissue sections. GL, glomerulus; PT, proximal tubule; LP, loop of Henle; DT, distal convoluted tubule; CD, collecting duct. Scale bar, 20 μm.

in Tg mice indicated urine Na$^+$ loss, leading to extracellular volume contraction, consistent with the increased BUN (Fig 2E). However, the body weights were similar in Tg (19.4 ± 1.0 g) and WT (19.0 ± 1.3 g) (S3 Fig) mice after the low salt diet, suggesting that the increased BUN may be related to protein breakdown, instead of dehydration.

## Vpr reduced NCC protein expression without change in MR protein expression in Tg mouse kidneys

Considering that NCC plays a major role in sodium homeostasis and that thiazide diuretics are the first drugs of choice in the treatment for uncomplicated hypertension, we next investigated whether NCC protein expression is altered in the Tg mice. Indeed, in Tg mice compared with WT mice, NCC protein expression was markedly decreased measured by immunoblotting (Fig 3A), immunohistochemistry (Fig 3C) and immunofluorescence microscopy (S1 Fig). However, the protein expressions of MR and Na$^+$/K$^+$-ATPase were not changed, as determined by immunoblotting of kidney homogenates from Tg and WT mice (Fig 3B). NCC expression and activity, mainly mediated by MR activity, are regulated by aldosterone [17,25]. We, therefore, measured plasma aldosterone concentration and renin activity. Levels of both were similar in Tg and WT mice, although those levels were significantly elevated after salt depletion in both groups (Fig 2F and 2G).

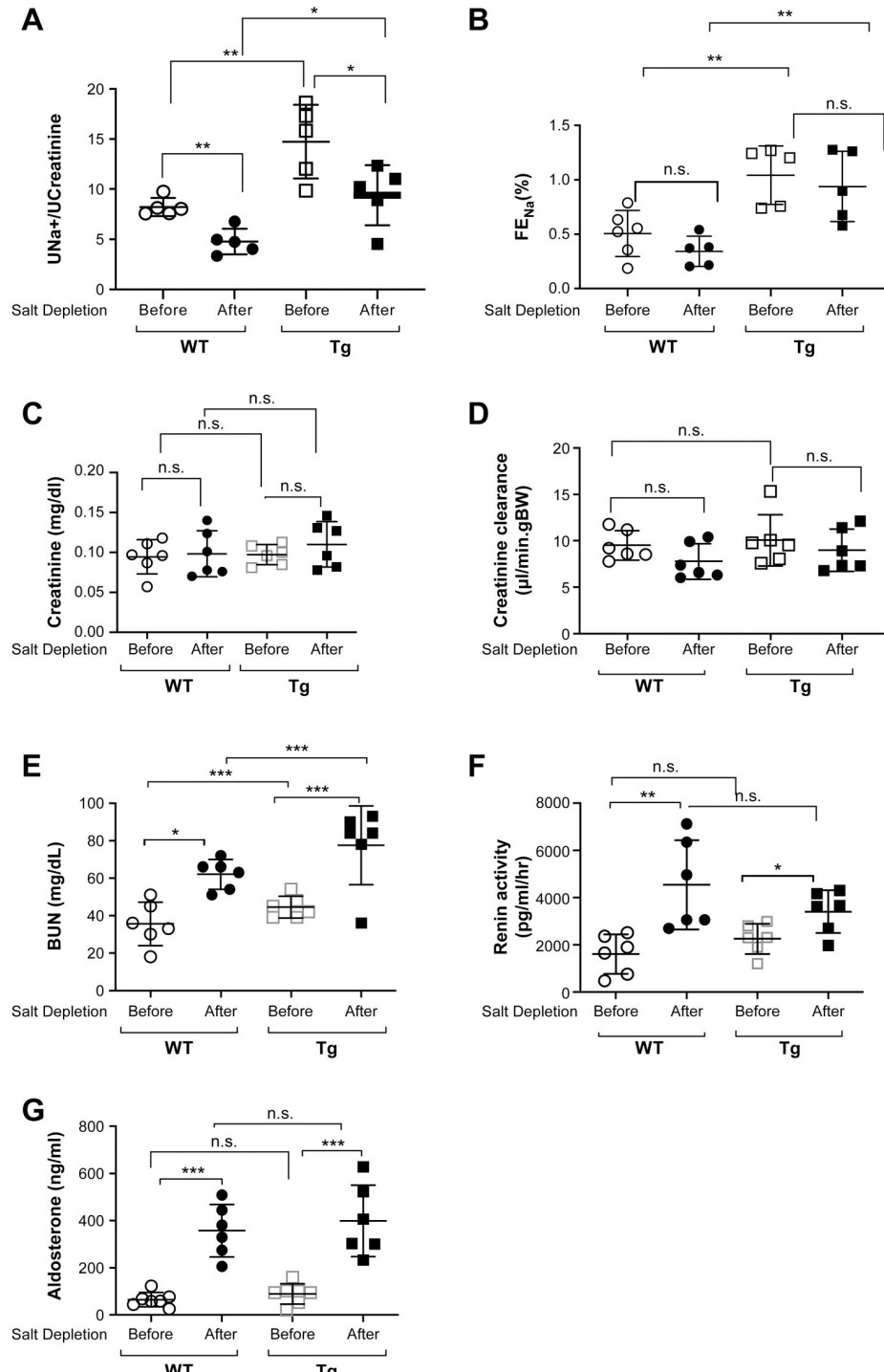

**Fig 2. Kidney function and sodium homeostasis-related parameters in Vpr Tg mice.** (**A**) Urinary sodium excretion (UNaV, ratio of urinary sodium in mmol/Lto urine creatinine in mmol/L) was significantly elevated in Tg mice compared with WT littermates, before and after feeding a low salt diet. (**B**) Fractional excretion of sodium (FENa) was higher in Tg compared with WT mice before and after a low salt diet. (**C**) Serum creatinine concentrations and (**D**) creatinine clearances (Ccr) were not different among the groups. (**E**) Blood urea nitrogen (BUN) concentration was higher in Tg that WT mice before and after allow salt diet. (**F**) Plasma renin activity and (**G**) plasma aldosterone concentrations were increased on low salt diet but were similar in WT and Tg mice. n = 5–6. n.s., not significant; $^*P < 0.05$, $^{**}P < 0.01$, $^{***}P < 0.001$, ANOVA, Bonferroni correction.

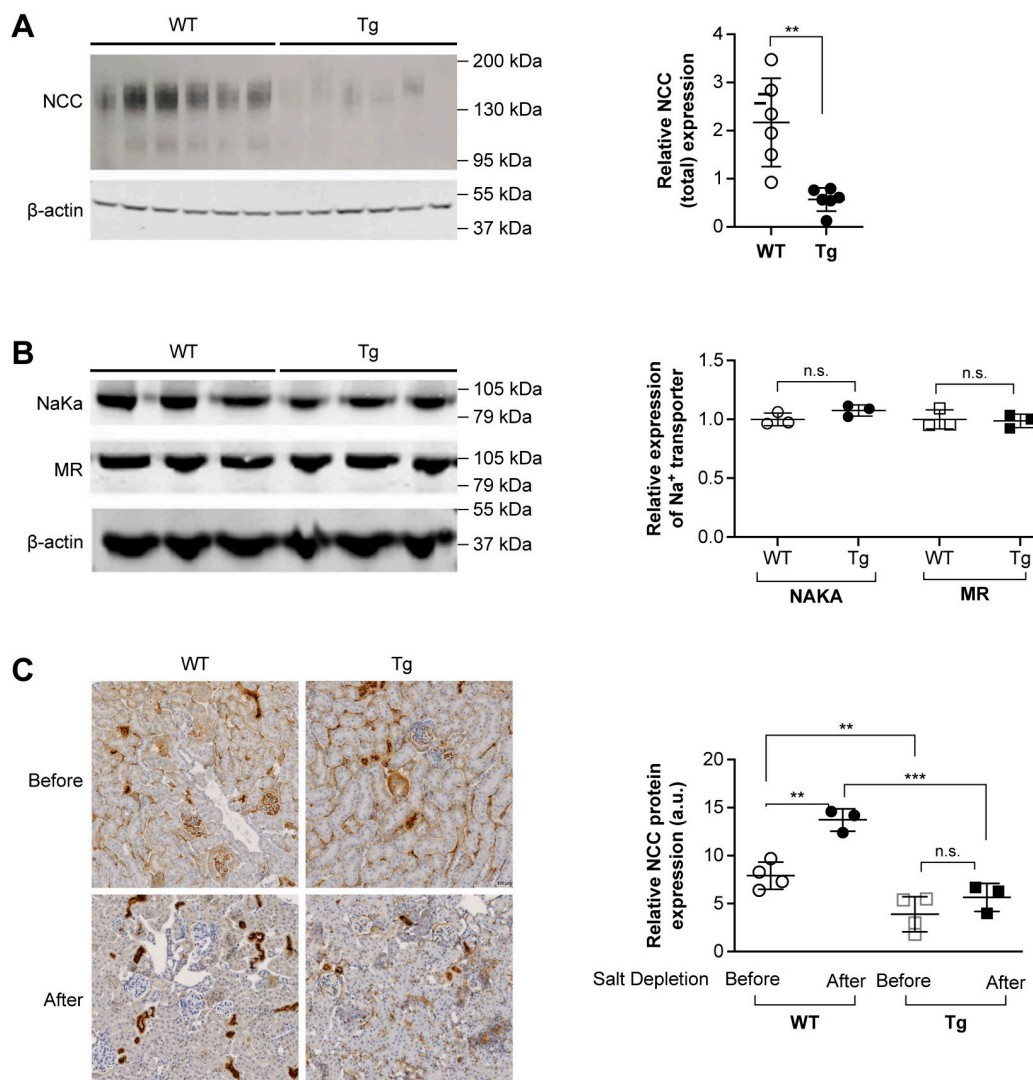

**Fig 3. Expression of NCC, NaKa, and MR in kidney homogenates from wild-type (WT) and Vpr transgenic (Tg) mice.**
(A) Protein expression of total NCC was markedly suppressed in kidneys from Tg mice compared with WT littermates.
n = 6. $^{**}P < 0.01$, Student's $t$ test. (**B**) Protein expressions of NaKa and MR were similar in WT and Tg mice. NaKa, Na$^+$/
K$^+$-ATPase. n = 3. n.s., not significant, Student's $t$ test. (**C**) Immunohistochemical analysis of NCC distribution in kidney
cortex sections demonstrated that NCC is located in the apical cellular regions of the renal distal convoluted tubule, with
similar distribution in both WT and Tg mice. NCC immunostained tubules were scanned and NCC protein expression was
found to be lower in Tg than WT mice both before and after salt depletion. n = 3–4. n.s., not significant; $^{**}P < 0.01$,
$^{***}P < 0.001$, ANOVA, Bonferroni correction.

## Vpr suppressed MR transcriptional activity

Vpr functions as a co-regulator of various nuclear steroid receptors [10,11,13] and, in particu-
lar, enhances GR transcriptional activity. As MR and GR share certain ligands and DNA-bind-
ing elements [26], we next investigated the effect of Vpr on the MR transcriptional activity by
*in vitro* transfection of African green monkey kidney (CV-1) cells, which lack endogenous MR
and GR [27], with plasmid pcDNA-hMR, encoding the human MR. In MR-transfected CV-1
cells, aldosterone markedly stimulated luciferase activity from the MR-responsive MMTV pro-
moter, while Vpr suppressed transcriptional activity in an aldosterone-dependent manner (Fig

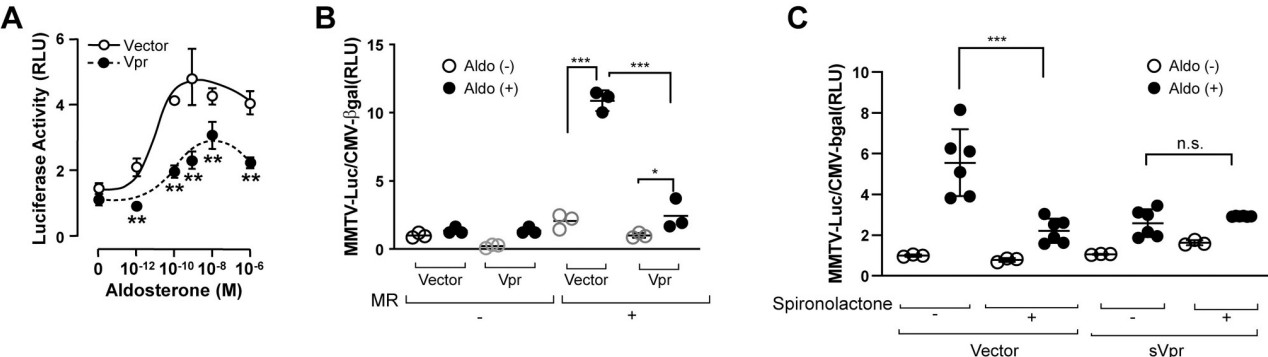

**Fig 4. Suppression of MR transcriptional activity by Vpr in African green monkey kidney (CV-1) cells.** (**A**) CV-1 cells, lacking MR, were transfected with Vpr- and MR-expressing plasmids. MR-mediated transcription activity, monitored by the MR-responsive MMTV-luciferase reporter, exhibited a sigmoidal dose-response to aldosterone and across the aldosterone concentration range, which was inhibited by Vpr. (**B**) Vpr markedly suppressed MR transcriptional activity in CV-1 cells transfected with MR and cultured with aldosterone (100 nM). (**C**) Both sVpr and MR antagonist spironolactone (1 μM) suppressed aldosterone-induced (100 nM) MR transcription activity; the spironolactone-mediated suppression was not further reduced by sVpr. n = 3–6. n.s., not significant; $^*P < 0.05$, $^{***}P < 0.001$, ANOVA, Bonferroni correction.

4A), but in the absence of MR, neither aldosterone nor Vpr affected promoter activity (Fig 4B).

As with the intracellular expression of Vpr following cDNA transfection, the addition of synthetic Vpr full length peptide (sVpr) to culture media also suppressed aldosterone-induced transcriptional activity of the MMTV promoter (Fig 4C). Treating the CV-1 cells, alone, with spironolactone (1 μM), an MR antagonist, abrogated the aldosterone-induced increase in MMTV transcriptional activity, and no additional suppressive effect was observed when sVpr and spironolactone were added together (Fig 4C). These results suggest that Vpr, both intracellularly expressed or exogenously added, functions as a potent corepressor of MR transcriptional activity.

In human DCT cells, Vpr suppressed the expression of endogenous *SLC12A3* gene and NCC protein, and addition of eplerenone, an MR antagonist, had no further suppressive effect (S4 Fig), indicating Vpr-mediated suppression of NCC mRNA and protein expression was through suppression of MR activity.

## Vpr suppressed NCC (*Slc12a3*) gene transcription

Vpr Tg mice manifested decreased NCC protein expression, and therefore, we investigated whether Vpr suppressed the *Slc12a3* (NCC) gene transcription. Indeed, the gene transcription of *Slc12a3* (Fig 5A), measured by qRT-PCR, was markedly reduced in Vpr Tg mice compared with WT mice. *In situ* hybridization experiments with RNAScope technology confirmed that the *Slc12a3* RNA expression was significantly decreased (fewer cells express *Slc12a3* RNA) in Tg mice compared with WT mice (Fig 5B).

These results suggest that Vpr suppresses transcription of *Slc12a3*, which accounts, at least in part, for the reduced NCC protein expression in Vpr Tg mouse kidneys.

## Vpr suppressed MR binding to the *SLC12A3* (NCC) promoter

We next investigated whether a functional MRE is present in human *SLC12A3* [24]. We searched for transcription factor binding motifs in the *SLC12A3* gene in the region from 5 kb upstream of the transcription start site to downstream of the third exon (~3kb). We identified putative transcription factor binding motifs in this region (S2 Table). Among these putative

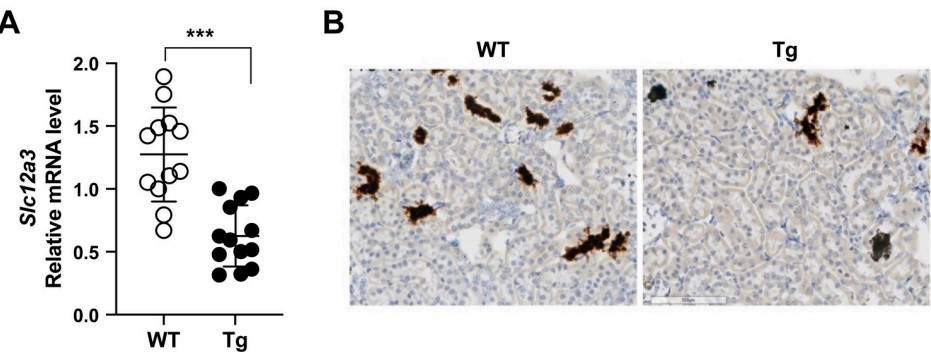

**Fig 5. Gene expression of *Slc12a3* in the kidney of wild-type (WT) and Vpr transgenic (Tg) mice. (A)** Renal *Slc12a3* gene expression, quantified by qRT-PCR, was significantly lower in Tg mice compared with WT mice. **(B)** *Slc12a3* gene expression was markedly lower (fewer cells express *Slc12a3*) in Tg mice compared with WT mice, as determined *in situ* hybridization, using RNAScope. n = 12–13. ***$P < 0.001$, Student's *t* test.

motifs, we identified seven amplicons (P-9 to P-15) that were significantly enriched in cells cultured with aldosterone, as compared with control cells (Fig 6A). Following exposure of human DCT cells to sVpr, the expressions of six aldosterone-responsive amplicons (P-9 to P-14) were attenuated (Fig 6A). Among these amplicons, P-10 contains an SP1 binding site, as well [24], suggesting that P-10 has the capacity to bind both MR and SP1.

Due to the close proximity of amplicons P-9 and P-10, which include putative MRE and SP1 motifs, respectively, to the transcriptional start site and the effect of sVpr to diminish aldosterone-induced enrichment, we studied this region by site-directed mutagenesis with random nucleotide sequences (S2 Table). As shown in (Fig 6B), for the wild type sequences, aldosterone induced a significant increase in transcriptional activity of the reporter construct, and this increase was markedly suppressed by sVpr. Single or dual mutation (mutated P-9 only, mutated P-10 only, or both mutated) prevented the aldosterone stimulation and abrogation of Vpr effects on the aldosterone-induced NCC reporter activity (Fig 6B). These results indicate that both the MRE and the SP1 binding motifs are required for MR transcription and for the suppressive effect of Vpr on *SLC12A3* transcription.

## Discussion

In this study, we generated doxycycline-inducible Vpr Tg mice and characterized renal Vpr expression. Urinary sodium excretion in Vpr Tg mice was markedly increased compared with WT littermates. Furthermore, Vpr suppressed MR-dependent transcriptional activity, by targeting the promoter region of *SLC12A3*, encoding the sodium-chloride transporter, located in the renal DCT. Vpr reduced renal NCC (*Slc12a3*) gene and protein expression, contributing to the increased urinary sodium excretion in Vpr Tg mice. These results are consistent with a recent observation that more than 80% of proteins suppressed by Vpr are nuclear proteins [28].

MR physiologically functions to promote gene transcription of transporters that promote transepithelial sodium transport [17,29–31]. NCC is mainly expressed in DCT1, and gradually decreases along DCT2 [32]. Chronic aldosterone administration increases renal NCC abundance and activity in DCT cells [29–31]. A putative MRE beginning at -156 position with regard to the transcriptional start site is present in the promoter of the human *SLC12A3* gene [24]. Indeed, the amplicon P-9 described here (caatcaaatggTGTTCTgc) was confirmed to be a functional MRE by site-directed mutagenesis analysis, while amplicon P-10

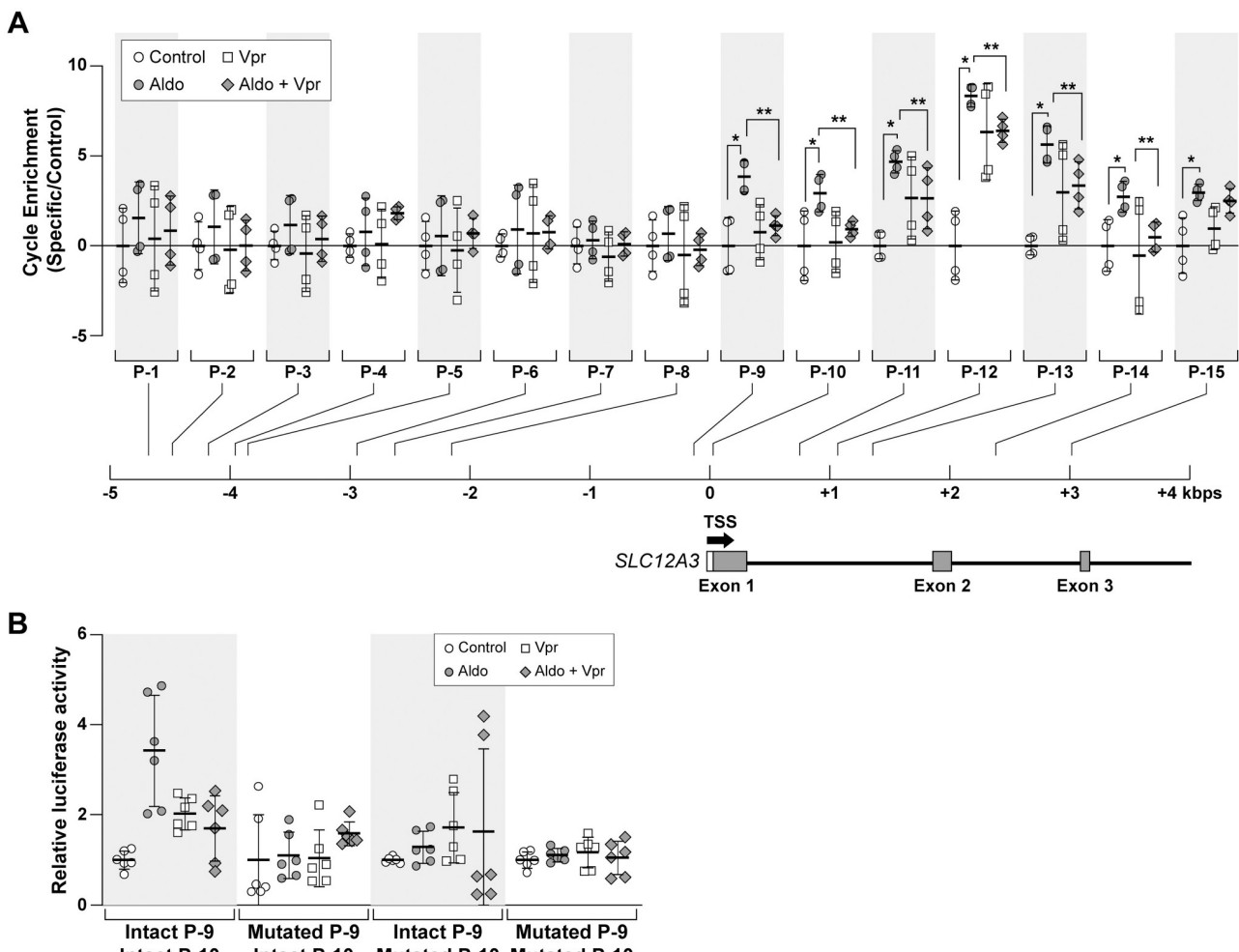

**Fig 6. Characterization of putative transcription factor motifs within human *SLC12A3* gene and the effect of sVpr on the association of MR to its binding regions.** (**A**) Human DCT cells were exposed to aldosterone or vehicle, with or without sVpr, and were processed for the ChIP assay, using anti-FLAG antibody, targeting FLAG-MR. Fifteen primer pairs (S2 Table), located around the transcription start site of the *SLC12A3* gene, encoding NCC, were used for qRT-PCR. The fold-enrichment of the specific FLAG-MR-binding DNA sequences to that of the normal rabbit anti serum control was compared among four treatment groups: Control, aldosterone, sVpr, and aldosterone plus sVpr. As shown in the column graphs, the regions of highest Vpr-specific binding (determined by the ratio of aldosterone response to Vpr response) were amplicons P-9 and P-10. $^{*}P < 0.05$ aldosterone vs control; $^{**}P < 0.05$ aldosterone plus sVpr vs. aldosterone. ANOVA, Bonferroni correction. (**B**) CV-1 cells were transfected with WT or mutated *SLC12A3* constructs. The construct pTSC-Luc contains one putative MR response element (MRE) motif (5'-CAATCAAATGGTGTTCTGC-3', amplicon P-9) and one putative SP1 motif (5'- CCCTCCCTGGACACC-3', amplicon P-10). Amplicons P-9 and P-10 exerted the largest suppressive effects by Vpr on transcriptional activity. Putative MRE and SP1 motifs flanking the *SLC12A3* transcription start site were mutated with random sequences (S2 Table) to eliminate the putative regulatory motif sequences. Each reporter construct contained a fragment of the *SLC12A3* gene. CV-1 cells were transfected with reporter constructs as shown and exposed to aldosterone and/or sVpr. sVpr attenuated aldosterone-stimulated reporter activity with the intact P-9 and P-10 constructs but had no effect with single or dual mutants of P-9 and P-10 amplicons. $^{***}P < 0.001$ vs control, ANOVA, followed by Bonferroni correction.

(CCCTCCCTGGacacc), a SP1-binding site at -28 position [24], was shown to be required for MR binding and proper function.

These results indicate that both MRE and SP1 motifs are required for its MR transcriptional activity and that Vpr can suppress MR transcriptional activity by interfering with either the MRE or the SP1 motif in the *SLC12A3* promoter. We also identified multiple MRE sites within introns 1 and 2 of the human *SLC12A3* gene. Previous work demonstrated that in some cases a particular MRE can bind to either MR or GR and therefore, can function both as an MRE and

a GRE response element [33]. However, the MRE at position -156 (amplicon P-9) appears specific for MR, because deletion of this site does not change the transcriptional activity of the *SLC12A3* promoter upon addition of deoxycorticosterone acetate [24]. Similarly, a unique MRE sequence has been demonstrated in the promoter of the intermediate conductance $K^+$ channel gene (*Kcnn4*) in rat distal colon [34].

The data presented here do not exclude the possibility that functional MREs in the *SLC12A3/Slc12a3* promoter may interact with the GR. Expression of the glucocorticoid-inactivating 11β-hydroxysteroid dehydrogenase type 2 (11βHSD2) in the distal nephron metabolizes glucocorticoids by converting cortisol/corticosterone to cortisone [35,36], which facilitates MRE/MR binding. In the experiments described here, Vpr Tg mice had increased urinary $Na^+$ excretion without an increase in glomerular filtration rate, the latter being a physiological and pathological effect of glucocorticoids [37]. In the present study, there was no change in creatinine clearance (reflecting glomerular filtration rate) in Vpr Tg mice, indicating that Vpr most likely does not function as a GRE regulator. As mentioned above, amplicon P-9 of *SLC12A3* is an aldosterone-responsive gene with a functional MRE, which suppressed MR transcriptional activity in the presence of Vpr. The MRE at position -156 (amplicon P-9) located in the *SLC12A3* promoter is specific for MR. Other potential MREs are specific for MRE or also binding to GRE needs further investigation. Reduction of endogenous *SLC12A3* gene expression by Vpr could be caused by suppression of MR activity in human DCT cells, as well (S4 Fig) The reduction of the MR/MRE interaction is a novel mechanism underlying the *SLC12A3/Slc12a3* gene repression which contributed to the increased urinary sodium excretion in Vpr Tg mice.

In this study, Vpr did not affect the plasma renin activity before and after salt depletion; in WT and Tg mice plasma renin activities were similar on ad libitum sodium diet and similarly increased by a low sodium diet. This is consistent with a recent clinical study showing that serum renin activities were similar in non-HIV and HIV-infected subjects while on ad libitum sodium diet and were similarly increased by a low sodium diet [38]. HIV enhances Ang II production and renin expression through the vitamin D receptor [39]; the role of vitamin D in the HIV-related renin response to a low sodium diet remains to be determined.

Aldosterone, synthesized in the adrenal cortex, is regulated by angiotensin II, $K^+$, and adrenocorticotrophic hormone. In the present study, plasma aldosterone concentrations were similar between WT and Tg mice, and increased with salt depletion in both groups, indicating that Vpr did not impair the aldosterone synthesis and regulation system in Tg mice. Aldosterone is the most important hormone in the regulation of $Na^+$ homeostasis in the distal nephron. One may argue that aldosterone regulates $Na^+$ homeostasis directly in the connecting tubule and the CD, and indirectly in DCT [40]. However, due to the technical difficulty, the role of aldosterone in DCT may have been underestimated [41].

The role of aldosterone in directly or indirectly regulating the renal tubular transport of $Na^+$ and other electrolytes, related to NCC remains controversial. Czogalla *et al.*, in a mouse model with random deletion of MR in 20% of renal tubule cells, found that upregulation of NCC phosphorylation in response to low NaCl diet is MR-independent [42]. Terker et al, in kidney-specific MR knockout mice, showed that the reduced NCC phosphorylation is $K^+$ dependent [43] and Cheng et al demonstrated that aldosterone increased NCC activity within minutes *ex vivo* and *in vivo* [44].

NCC is known to be regulated at the transcriptional and also at the posttranslational level, through phosphorylation, ubiquitination, and glycosylation by the renin-angiotensin system, $K^+$, kinases, and hormones [45,46]. Aldosterone regulation of NCC expression and activity is consistent with the expression of 11βHSD2 at the DCT segment [35,36]. Aldosterone is regulated by $K^+$ and *vice versa*.

It might be argued that Vpr-mediated reduction of NCC expression could be an indirect effect of the increase in $K^+$ concentration. However, our data showed that the plasma renin activity (Fig 2) and serum $K^+$ (S2 Fig) levels were similar between the WT and Tg mice both before and after salt depletion. Whether or not transgenic Vpr expression alters the aldosterone/MR-mediated gene and/or protein expression of sodium transporters and exchangers (*e. g.*, $Na^+$-$K^+$-$2Cl^-$ cotransporter/NKCC2 and epithelial sodium channels/ENaCs) along the nephron segment requires further investigation.

HIV-1 infection induces a host cellular metabolic changes, including those involved in glucose, lipid, and energy metabolism. The resulting shift from oxidative phosphorylation to aerobic glycolysis is important for maintaining retroviral quality and infectivity [47]. Vpr modulates host adipocyte [11], hepatocyte [13], and macrophage [48] metabolic pathways and their bioenergetics. Vpr induces PPAR β/δ transcriptional activity, and increases phosphorylated PDH subunit E1α, leading to decreased activity of the PDH [13], an enzyme complex that plays a critical role in sodium transport in the renal tubule and contributes to the pathogenesis of hypertension in spontaneously hypertensive rats [14]. Sodium transport in all nephron segments depends on both cytosolic glycolysis [49] and mitochondrial oxidative phosphorylation [50]. New proteomic and single cell RNA-seq techniques [51] will further clarify whether Vpr, as a co-regulator of MR or PPAR, reduces the expression and activities of PDH complex and other metabolic enzymes leading to the decrease in sodium reabsorption in the distal tubule.

A major consequence of HIV infection is the decrease in $CD4^+$ T cells and other immune and nonimmune cells. Vpr, released from infected cells, is sufficient to induce G2/M cell cycle arrest and apoptosis in bystander cells, including renal tubule cells [2,52]. In the renal DCT, loss of NCC accompanies DCT atrophy, downregulation of DCT-specific $Mg^{2+}$ channel TRPM6 and increase in cleavage of epithelial $Na^+$ channel [53]. In addition to MRE-mediated suppression of *Slc12a3* gene expression, further investigation is required to determine the mechanisms responsible for the reduction in NCC expression. Vpr directly or indirectly induces NCC degradation, upregulates specific miRNA [54] to suppress expression of *Slc12a3* gene (encoding NCC), modulates cellular apoptosis pathways to favor retention of retroviral over host proteins [55], and induces DCT atrophy in Vpr Tg mice.

Limitations of this study are acknowledged. In *Pepck1* promoter-driven Vpr Tg mice, Vpr is expressed in liver, kidney, and adipocytes and is present in the plasma [22]. Endogenous PEPCK1 is predominantly expressed in the renal proximal tubule and its expression in other nephron segments is low [51,56]. However, the *Pepck1* promoter (containing nt -2,086 to +69 in this study)-driven Vpr, integrated into the mouse genome, resulted in a difference in Vpr expression from that of endogenous *Pepck1* expression and also variable Vpr expression among mice [22]. In the current study, Vpr RNA was expressed along all nephron segments, albeit low level expression, which is consistent with previous observations that low level of Vpr expression causes many pathological effects in the host. Although endogenous cytosolic Pepck1 is predominantly expressed in the renal proximal tubule, low levels of *Pepck1* are expressed in the DCT cluster, as observed in the data from single-nucleus RNAseq [57].

Lifton [58,59] and colleagues reported that dietary salt intake is increased in patients with Gitelman syndrome compared with control subjects, suggesting that salt wasting caused by NCC deficiency induces a compensatory increase in salt consumption. In the present study, loss of NCC expression and function in Vpr Tg mice was associated with increased urinary sodium excretion. However, similar serum $Na^+$ concentrations, urine excretions, and body weights (S3 Fig) were similar in Vpr Tg and WT littermates. Whether or not the maintenance of these parameters in Vpr Tg mice to the levels seen in WT littermates could not be

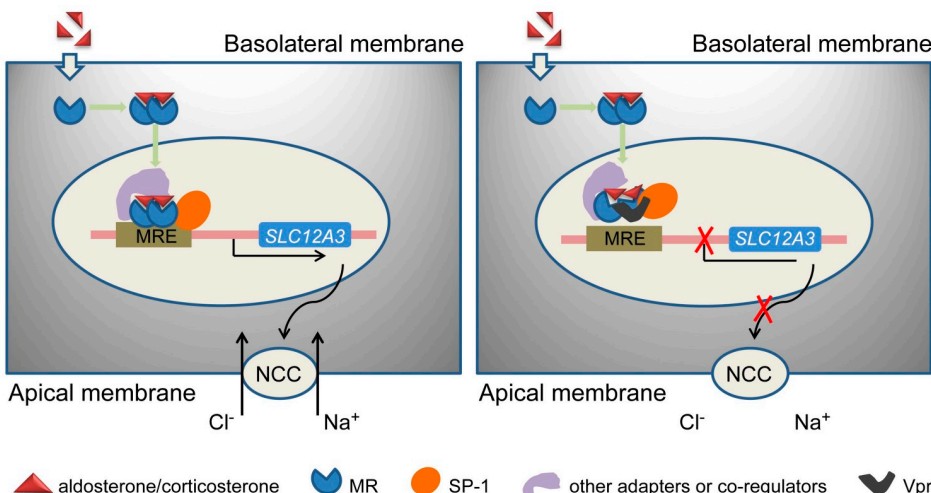

**Fig 7. A schematic model of Vpr suppression of *SLC12A3* transcription activity in the distal convoluted tubule.**
(**A**) In the absence of Vpr, aldosterone binds MR in the cytoplasm. The activated MR translocates to the nucleus, where it binds the mineralocorticoid response element (MRE) in genomic DNA. This complex, together with SP1 and other transcription factor(s) and other co-regulator(s), activates the transcription initiation complex and thereby promotes *SLC12A3* gene transcription and NCC protein translation. Although both MR (aldosterone) and GR (cortisol, corticosterone etc.) hormones potentially bind to MRE, the presence of 11βHSD2 in the DCT, especially DCT2, converts cortisol (humans)/corticosterone (rodents) to cortisone, which facilitates MR binding to MRE. (**B**) Vpr, binding either MR or SP1 and other transcription factors, abrogates the interaction of MR and MRE and thus prevents *SLC12A3* transcription and translation. This reduces the abundance of NCC in the apical membranes of distal convoluted tubules. Considering that plasma aldosterone levels in Tg mice were similar to those of WT mice, and aldosterone concentrations were increased by salt depletion in both groups, Vpr did not impair aldosterone synthesis and regulation system in Tg mice. Of note, MREs can be located within intron or other regions of the gene, although it is illustrated here in the promoter region of *SLC12A3*.

determined because we did not record the salt and food intake of mice during the study period, a limitation of our current study.

## Conclusion

In conclusion, Vpr Tg mice manifested reduced MR transcriptional activity, which reduced gene expression of *SLC12A3* (encoding the thiazide-sensitive sodium cotransporter, NCC) and thereby decreased the abundance of NCC protein in the apical membrane of the distal nephron (Fig 7). Less NCC protein facilitated increased urinary sodium excretion in these mice, due to reduced tubular sodium reabsorption. The increase blood urea nitrogen in Vpr Tg mice could be a result of intravascular volume depletion. However, body weight was not perturbed. These data suggest a pathological mechanism that may contribute to sodium loss in HIV-positive patients that could be compensated by increased sodium intake.

## Supporting information

**S1 Fig. NCC distribution in kidney sections from wild-type (WT) and Vpr transgenic (Tg) mice.** Mouse kidney sections were stained using NCC (green) or Vpr (red). NCC is expressed in the apical membrane of distal convoluted tubule in both WT and Tg mice, whereas Vpr is expressed in tubules in Vpr Tg mice but not in WT mice. Nuclei were stained in blue with Hoechst 33342. Magnification x300.
(TIF)

**S2 Fig. Concentrations of major plasma electrolytes and uric acid of Tg and WT mice fed salt depletion diets.** (**A**) $Na^+$, (**B**) $K^+$, (**C**) $Mg^{2+}$, (**D**) $Ca^{2+}$, and (**E**) $Cl^-$ concentration had no

difference between Tg and WT mice. n = 5. n.s., not significant, Student's *t* test. The electrolytes were measured at NIH Clinical Center Clinical Laboratory (Bethesda, MD).
(TIF)

**S3 Fig. Body weight and urine volume of WT and Vpr Tg mice.** (**A**) Four days after the low salt diet the mice were weighed. n = 7–10. n.s., not significant, Student's *t* test. (**B**) Mouse were placed in metabolic cages and overnight urine was collected before and after the mice were placed on low salt diet. n = 9. n.s., not significant, ANOVA, Bonferroni correction.
(TIF)

**S4 Fig. Suppression of MR transcriptional activity by Vpr in human DCT cells.** (**A**) Human DCT cells were treated with or without soluble Vpr (sVpr, 100ng/ml) for 24-hr followed by aldosterone treatment for the time indicated. Protein expression of NCC and MR were determined by immunoblotting. (**B**) The density of NCC was quantified by densitometry. The aldosterone-enhanced NCC protein expression, with a peak at 3-hr treatment, was abolished by sVpr (100ng/ml). n = 3. $^*P < 0.05$, $^{**}P < 0.01$. (**C**) Human DCT cells were treated with sVpr, aldosterone, or eplerenone (10μM), individually or in combination as indicated. Expression of *SLC12A3* mRNA was determined by qRT-PCR and normalized to β-actin mRNA. The aldosterone-induced increase in *SLC12A3* mRNA expression was attenuated by Vpr, but eplerenone had no further effect on *SLC12A3* mRNA expression. n = 4–6. $^{**}P < 0.01$, $^{***}P < 0.001$, ANOVA, Bonferroni correction.
(TIF)

**S5 Fig. NCC protein expression was expressed in the immortalized human renal distal convoluted tubule cells.** The immortalized human renal distal convoluted tubule cells were previously characterized [8]. Here, additional immunofluorescence staining was performed. Green, anti-NCC antibody (Cat. No.: AB3553, 1:500 dilution); Blue, nucleus. Scale bar, 100 μm.
(TIF)

**S1 Table. Potential MRE sequences within the *SLC12A3* promoter (-5kb to 3kb) and their detailed matrix.**
(DOCX)

**S2 Table. Target motifs and primer pair sequences for characterization of MRE sequences within human *SLC12A3* gene.**
(DOCX)

**S3 Table. Primary antibodies used in this study.**
(DOCX)

**S4 Table. Primers for qRT-PCR in mouse kidneys and hDCT cells.**
(DOCX)

## Acknowledgments

We appreciate assistance from Jurgen Heymann, Kris Ylaya, Ryan G Morris and Abeer Fadda.

## Author Contributions

**Conceptualization:** Huiyan Lu, Hidefumi Wakashin.

**Data curation:** James L. T. Dalgleish, Peng Xu, Laureano D. Asico, Joon-Yong Chung.

**Investigation:** Koji Okamoto, Huiyan Lu, Teruhiko Yoshida, Avi Z. Rosenberg.

**Resources:** Stephen Hewitt, John J. Gildea, Robin A. Felder.

**Visualization:** Khun Zaw Latt, Erik H. Koritzinsky.

**Writing – original draft:** Shashi Shrivastav, Hewang Lee.

**Writing – review & editing:** Pedro A. Jose, Mark A. Knepper, Tomoshige Kino, Jeffrey B. Kopp.

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
