## [Decision Letter · Decision Letter 0]

30 May 2022

PONE-D-22-10698HIV-1 Vpr suppresses expression of the thiazide-sensitive sodium chloride co-transporter in the distal convoluted tubulePLOS ONE

Dear Dr. Kopp,

Thank you for submitting your manuscript to PLOS ONE. After careful consideration, we feel that it has merit but does not fully meet PLOS ONE’s publication criteria as it currently stands. Therefore, we invite you to submit a revised version of the manuscript that addresses the points raised during the review process.

 Your manuscript was reviewed by two experts. Although they found your work interesting, they raised some concerns. Please revise it according to their suggestions. In addition, please consider performing genome-editing analyses as written in  the "Additional Editor Comments" section below.

We look forward to receiving your revised manuscript.

Kind regards,

Hodaka Fujii, M.D., Ph.D.

Academic Editor

PLOS ONE

Journal Requirements:

4.Thank you for stating the following in the Acknowledgments Section of your manuscript: 

"This work was funded by the NIDDK Intramural Research Program. We appreciate assistance from Jurgen Heymann, Kris Ylaya, Ryan G Morris and Abeer Fadda."

"This work was funded by the NIDDK Intramural Research Program."

"Author JBK holds a patent relating to monoclonal antibodies to HIV-1 Vpr and methods of using same.   United States Patent 7,993,647 (2015). No other conflicts of interest, financial or otherwise, are declared by the authors."

6. PLOS ONE now requires that submissions reporting blots or gels include original, uncropped blot/gel image data as a supplement or in a public repository. This is in addition to complying with our image preparation guidelines described at https://journals.plos.org/plosone/s/figures#loc-blot-and-gel-reporting-requirements. These requirements apply both to the main figures and to cropped blot/gel images included in Supporting Information. If the manuscript is positively reviewed, we will ask the authors to provide any missing raw image data for blot/gel results when they submit their first revision. As part of your review, please ensure that figures reporting blot or gel images comply with the journal’s image preparation guidelines and that the original data are provided following the journal’s request.  If you have any questions or concerns about blot/gel figures or data for this submission, please email us at plosone@plos.org before issuing a decision letter.

7. We note that you have included the phrase “data not shown” in your manuscript. Unfortunately, this does not meet our data sharing requirements. PLOS does not permit references to inaccessible data. We require that authors provide all relevant data within the paper, Supporting Information files, or in an acceptable, public repository. Please add a citation to support this phrase or upload the data that corresponds with these findings to a stable repository (such as Figshare or Dryad) and provide and URLs, DOIs, or accession numbers that may be used to access these data. Or, if the data are not a core part of the research being presented in your study, we ask that you remove the phrase that refers to these data.

Additional Editor Comments:

It would be preferable to mutagenize MRE and SP1 motifs in the endogenous gene promoter using genome editing techniques and examine expression levels of the SLC12A transcript.

Reviewers' comments:

Reviewer's Responses to Questions

**Comments to the Author**

1. Is the manuscript technically sound, and do the data support the conclusions?

Reviewer #1: Yes

Reviewer #2: Yes

2. Has the statistical analysis been performed appropriately and rigorously? 

Reviewer #1: Yes

Reviewer #2: Yes

3. Have the authors made all data underlying the findings in their manuscript fully available?

Reviewer #1: Yes

Reviewer #2: Yes

4. Is the manuscript presented in an intelligible fashion and written in standard English?

Reviewer #1: Yes

Reviewer #2: Yes

5. Review Comments to the Author

Reviewer #1: Vpr-induced salt wasting and loss of NCC protein expression in mouse kidney was well studied here and, in parallel, Vpr-induced decrease in genomic bindings of MR to Slc12a3 gene location and that in the aldosterone-dependent transcriptional activity were also well demonstrated in renal cell-lines. However, considering the difficulty in detecting a change of MR genomic bindings in in-vivo DCT, the author’s hypothesis would have to be more supported by other data which associates these two parts of the study closely.

Main concerns:

1. Not only the genomic bindings of MR by Vpr treatment in human DCT cell-line, was the endogenous expression level of Slc12a3 also decreased ? At least, such changes in expression level of NCC protein and/or mRNA should be demonstrated.

2. As shown in Fig 3a, NCC protein level was clearly diminished in the kidney of Vpr-transgenic mice compared to that of wild-type mice. However, the intensity of signal in each of positive-cells in IHC and in-situ hybridization as shown in Fig. 3c and 5b seems strong almost equally in both of the groups of mice, leading to a doubt that the Vpr-induced change in renal NCC protein level would not be derived from the cellular expression level of Slc12a3 but the number of DCT cells in the kidney. At least, it should be shown that the number of DCT cells in each of the sections were not affected with the transgenic expression of Vpr.

3. About the supplemental Fig 3, the staining pattern of NCC seems rather cytoplasmic, which does not indicate strong expression of NCC in plasma membrane. More conservative statement would be more acceptable for readers.

Minor concern:

1. Some typos are observed, such as doubling of “PMC” in the reference section.

Reviewer #2: The work is very well done, experiments are well controlled and supporting evidence from various sources is nicely used. The work clearly supports an effect of VPR in the renal tubules, specifically upon MR and then NCC. My main critique of the manuscript is that this observed effect upon NCC may not be the entirety or even the major mechanism by which HIV/VPR mediated sodium wasting occurs.

Questions:

Why aren’t aldo and renin levels increased compared to controls?

Mice were fed the same chow, but was intake controlled or did actual intake vary among the mice (I expect so but not explicitly stated)? In a study of Gitelman’s patients, Lifton famously found that sodium intake among affected subjects (and heterozygotes) increased compared to unaffected individuals. Is it possible that the low sodium chow designed to induce increased aldo levels prevented us from seeing a further physiologic rise in renin/aldo? Does this in turn prevent the usual compensatory increases in sodium reabsorption along the nephron (ENaC specifically)?

I agree with the investigators that “Whether or not transgenic expresssion alters the

aldosterone/MR-mediated gene and/or protein expression of sodium transporters and exchangers

(e.g. Na+-K+-2Cl- cotransporter/NKCC2 and epithelial sodium channels/ENaCs) along the

nephron segment requires further investigation.” While the component of NCC in this phenomenon could be examined with a series of laborious thiazide treatments, this is likely to be of limited scientific value to the human disease studied due to differences in tubule contributions to sodium reabsorption across species. I would be sure to caveat my conclusions to indicate that possibility that NCC only contributes to this effect.

6. PLOS authors have the option to publish the peer review history of their article (what does this mean?). If published, this will include your full peer review and any attached files.

Reviewer #1: **Yes: **Ueda Kohei

Reviewer #2: No

---

## [Author Response · Author response to Decision Letter 0]

13 Jul 2022

We appreciate the comments of the editor and reviewers, which helped us to further improve the manuscript. Our point-by-point answers to the reviewers’ concerns are detailed in the separate file “Response to reviewers” of our resubmission.

Our response to Journal requirements #3 (Funding Information): This work was funded by the NIDDK Intramural Research Program 1ZIADK043411-15.

Our response to Journal requirements #5 (Competing Interests): The statement: "This does not alter our adherence to PLOS ONE policies on sharing data and materials.” has been added in the “Disclosure statement” section of the revised manuscript.

Further responses to the reviewer's and editor's comments are provided in the submitted document "Response to reviewers_07132022.docx"

---

## [Decision Letter · Decision Letter 1]

8 Aug 2022

HIV-1 Vpr suppresses expression of the thiazide-sensitive sodium chloride co-transporter in the distal convoluted tubule

PONE-D-22-10698R1

Dear Dr. Kopp,

We’re pleased to inform you that your manuscript has been judged scientifically suitable for publication and will be formally accepted for publication once it meets all outstanding technical requirements.

Kind regards,

Hodaka Fujii, M.D., Ph.D.

Academic Editor

PLOS ONE

Additional Editor Comments (optional):

Reviewers' comments:

Reviewer's Responses to Questions

**Comments to the Author**

1. If the authors have adequately addressed your comments raised in a previous round of review and you feel that this manuscript is now acceptable for publication, you may indicate that here to bypass the “Comments to the Author” section, enter your conflict of interest statement in the “Confidential to Editor” section, and submit your "Accept" recommendation.

Reviewer #1: All comments have been addressed

Reviewer #2: All comments have been addressed

2. Is the manuscript technically sound, and do the data support the conclusions?

Reviewer #1: Yes

Reviewer #2: (No Response)

3. Has the statistical analysis been performed appropriately and rigorously? 

Reviewer #1: Yes

Reviewer #2: (No Response)

4. Have the authors made all data underlying the findings in their manuscript fully available?

Reviewer #1: Yes

Reviewer #2: (No Response)

5. Is the manuscript presented in an intelligible fashion and written in standard English?

Reviewer #1: Yes

Reviewer #2: (No Response)

6. Review Comments to the Author

Reviewer #1: The authors added as much data as possible to support their hypothesis and corrected their statement to be more acceptable for readers.

Reviewer #2: (No Response)

7. PLOS authors have the option to publish the peer review history of their article (what does this mean?). If published, this will include your full peer review and any attached files.

Reviewer #1: No

Reviewer #2: No

---

## [Editor Report · Acceptance letter]

1 Sep 2022

PONE-D-22-10698R1 

HIV-1 Vpr suppresses expression of the thiazide-sensitive sodium chloride co-transporter in the distal convoluted tubule 

Dear Dr. Kopp:

I'm pleased to inform you that your manuscript has been deemed suitable for publication in PLOS ONE. Congratulations! Your manuscript is now with our production department. 

Kind regards, 

on behalf of

Dr. Hodaka Fujii 

Academic Editor

PLOS ONE